# Emerging Mechanisms of Endocytosis in *Toxoplasma gondii*

**DOI:** 10.3390/life11020084

**Published:** 2021-01-25

**Authors:** Olivia L. McGovern, Yolanda Rivera-Cuevas, Vern B. Carruthers

**Affiliations:** Department of Microbiology and Immunology, University of Michigan School of Medicine, Ann Arbor, MI 48109-5620, USA; livmcgov@umich.edu (O.L.M.); ryoly@umich.edu (Y.R.-C.)

**Keywords:** *Toxoplasma gondii*, *Plasmodium*, malaria, endocytosis, ingestion, intracellular trafficking

## Abstract

Eukaryotes critically rely on endocytosis of autologous and heterologous material to maintain homeostasis and to proliferate. Although mechanisms of endocytosis have been extensively identified in mammalian and plant systems along with model systems including budding yeast, relatively little is known about endocytosis in protozoan parasites including those belonging to the phylum Apicomplexa. Whereas it has been long established that the apicomplexan agents of malaria (*Plasmodium* spp.) internalize and degrade hemoglobin from infected red blood cells to acquire amino acids for growth, that the related and pervasive parasite *Toxoplasma gondii* has a functional and active endocytic system was only recently discovered. Here we discuss emerging and hypothesized mechanisms of endocytosis in *Toxoplasma gondii* with reference to model systems and malaria parasites. Establishing a framework for potential mechanisms of endocytosis in *Toxoplasma gondii* will help guide future research aimed at defining the molecular basis and biological relevance of endocytosis in this tractable and versatile parasite.

## 1. Introduction

Endocytosis of extracellular material is fundamental to eukaryotic organisms. Eukaryotic cells require endocytosis to maintain membrane homeostasis, turnover membrane proteins, downregulate receptors, and acquire exogenous nutrients, among other functions [1]. Although it is generally expected that all eukaryotes are capable of endocytosis, the existence of a functional endocytic pathway in the protozoan parasite *Toxoplasma gondii* remained unclear until recently. There were several reasons for this. First, *Toxoplasma gondii* resides in a membrane-bound intracellular compartment called the parasitophorous vacuole (PV) that in most cell types does not receive host material via vesicular fusion. Therefore, it was not clear whether *Toxoplasma gondii* could access host resources via endocytosis. Second, the existence of a complete set of endocytic organelles in *Toxoplasma gondii* was not appreciated until the discovery of the parasite lysosome termed the Plant-Like Vacuole or Vacuolar Compartment (VAC, the term used hereafter) [2,3]. Finally, while *Toxoplasma gondii* has the basic elements for endocytosis including endosome-like compartments (ELCs) marked by Rab5 or Rab7, several studies indicated that ELCs were repurposed as a conduit for exocytic trafficking en route to regulated secretory organelles (micronemes and rhoptries) involved in parasite invasion [4,5,6,7,8]. Together with earlier suggestive studies [9,10], more recent work has established that *Toxoplasma gondii* undergoes active endocytosis of host cytosolic material during intracellular infection via the so-called ingestion pathway [11].

During invasion, *Toxoplasma gondii* burrows into the host cell, pushing the host plasma membrane in toward the cytosol, and pinching it off to form its intracellular niche, the PV. This cloak of host-derived membrane provides a protective barrier that resists fusion with host lysosomes and limits autophagic destruction of the parasite [12,13,14,15]. However, the PV also represents a potential impediment to accessing nutrients in the host cell cytosol including amino acids, lipids and nucleotides [16,17]. This potential barrier is overcome in several different ways including the permeability of the PV to small molecules [18,19], sequestration of host organelles within the PV [20,21,22], and by endocytosis [11,23], which is the main focus of this review. *Toxoplasma gondii* endocytosis can be broken down into two main steps. First, material from the host cell cytosol is taken up across the PV membrane (PVM) and parasite plasma membrane into endocytic vesicles in the parasite cytosol. Second, endocytic cargo is delivered via the endosomal system to the VAC for degradation. This review will compare emerging insights into endocytosis in *Toxoplasma gondii* with established paradigms in model organisms and malaria parasites to illustrate the vast potential for discovery in the field. We will also discuss the prospects for discovering novel mechanisms of endocytosis given the peculiarities of *Toxoplasma gondii*. For an additional perspective on the topic, readers are referred to another recent review comparing endocytosis in malaria parasites and *Toxoplasma gondii* [24].

## 2. Endocytosis and Endolysosomal Trafficking in Model Organisms

### 2.1. Clathrin-Dependent Mechanisms of Endocytosis in Model Organisms

Endocytosis involves bending of the plasma membrane in toward the cytosol and scission of the plasma membrane to form endocytic vesicles. Mechanisms underlying this process in model systems can be divided into three categories—clathrin-dependent, clathrin-independent/dynamin-dependent, and clathrin- and dynamin-independent (Figure 1). These pathways will be discussed in turn, beginning with clathrin-mediated endocytosis (see also [1,25,26,27] for comprehensive reviews).

In mammalian cells, clathrin-mediated endocytosis facilitates internalization of transmembrane receptors. First, a nucleation complex, consisting of the F-BAR (Fes/CIP4 homology Bin-Amphiphysin-Rvs) domain proteins FCHO1/2 (Fer/Cip4 homology domain only 1 and 2), the clathrin adaptor complex for endocytosis adaptor protein 2 (AP-2), and the scaffolding proteins epidermal growth factor receptor substrate 15 (EPS15), EPS15-related protein (EPS15R) and intersectins 1 and 2 are recruited to phosphatidylinositol 4,5 bisphosphate (PI(4,5)P_2_ or PIP_2_) at the plasma membrane [1,27]. AP-2 serves as a major hub that binds to cytosolic domains of transmembrane cargoes, recruits accessory adaptors such as AP180/CALM (adaptor protein 180/clathrin assembly lymphoid myeloid), epsins and HIP1R (Huntingtin-interacting protein 1 related) and their associated cargoes and, together with the accessory adaptors, recruits clathrin to form the clathrin coat [1,27]. Membrane bending is driven by binding of BAR domains in FCHO1/2, insertion of amphipathic helices in membrane-binding ENTH (Epsin N-Terminal Homology Domain) and ANTH (AP180 N-Terminal Homology Domain) domains of epsin and CALM, respectively, and polymerization of a curved clathrin lattice that encases the forming vesicle, which may drive and/or stabilize membrane curvature [1,27]. Actin nucleation and branching-promoting proteins N-WASP (Neural Wiskott-Aldrich syndrome protein) and actin related protein 2//3 complex (Arp2/3) are also recruited, and drive polymerization of actin coupled to the clathrin coat through interactions with epsin and HIP1R. This is thought to push against the plasma membrane, elongating the neck of the forming vesicle and further promote membrane bending through tensile pressure [27,28,29]. Vesicle scission is mediated by recruitment of the N-BAR proteins endophilin and amphiphysin, which are thought to restrict the vesicle neck and also recruit the guanosine triphosphatase (GTPase) dynamin [27,30]. Dynamin is thought to act as a “pinchase” that forms a coil around vesicle necks and hydrolyzes guanosine triphosphate (GTP) to squeeze and sever the membrane to form endocytic vesicles [31]. Finally, the vesicle is uncoated and clathrin coat components are recycled back to the cytosol by HSC70 (heat shock cognate protein 70) and its cofactor auxillin/GAK (cyclin G-associated kinase) [1,27].

Although clathrin-mediated endocytosis is generally a highly conserved process, the requirement of key players in mammalian cells such as dynamin, AP-2 and other components of the nucleation complex varies among other eukaryotes. For example, the protozoan parasite *Trypanosoma brucei* undergoes clathrin-mediated endocytosis despite lacking AP-2 and having a single dynamin that is devoted to mitochondrial maintenance, but not endocytosis [32,33]. Plants lack FCHO1/2 and are not dependent on AP-2, but they have evolved a unique nucleator complex for clathrin-mediated endocytosis called the TPLATE complex, which is recruited to sites of clathrin-mediated endocytosis prior to AP-2 [34,35]. Similarly, AP-2 is not essential for yeast clathrin-mediated endocytosis; its only demonstrated role in clathrin-mediated endocytosis is for uptake of the yeast toxin K28 [1,36,37,38]. Finally, although the yeast dynamin Vps1 (vacuolar protein sorting 1) is required for efficient endocytic vesicle scission, its role in endocytosis is controversial. On the one hand, one study showed that Vps1 is recruited to sites of clathrin-mediated endocytosis, a dominant-negative Vps1 mutant severely inhibits endocytosis, and that Vps1 is able to tubulate liposomes in vitro, suggesting that dynamin may act as a pinchase, as in mammalian cells [39]. On the other hand, 22% of clathrin-mediated endocytic sites in this study lacked Vps1, and other studies have suggested that dynamin indirectly mediates membrane scission through its role in actin organization [39,40,41]. Accordingly, in less well understood systems, it should not be assumed that dynamin is necessary for clathrin-mediated endocytosis.

### 2.2. Clathrin-Independent Mechanisms of Endocytosis in Model Organisms

Mechanisms of clathrin-independent endocytosis are generally less understood and have been characterized mostly in mammalian cells. These pathways are diverse, but share common features, which include local production of phosphatidylinositols, and recruitment of Rho family GTPases (RhoA, Rac1 and Cdc42), actin nucleation proteins, and in some cases BAR domain proteins to promote membrane bending and scission for endocytic vesicle formation (see [42,43] for comprehensive reviews). For the purposes of this review, the FEME and CLIC-GEEC pathways will be described as representative examples of clathrin-independent endocytic pathways that are dynamin dependent or dynamin independent, respectively. 

Fast endophilin-mediated endocytosis (FEME) mediates endocytosis of activated G protein-coupled receptors. Lamellipodin binds to phosphatidylinositol 3,4 bisphosphate (PI(3,4)P_2_) at the plasma membrane and recruits the N-BAR protein endophilin. Endophilin is thought to induce membrane curvature and recruits N-WASP, Arp2/3, and dynamin to drive actin polymerization and endocytic vesicle formation [43]. This process is regulated by the GTPases Rac1 and RhoA and the kinase Pak1, which also likely mediate N-WASP and Arp2/3 recruitment [44]. Finally, dynamin is required for scission of endocytic vesicles in FEME [44].

CLIC/GEEC, on the other hand, is a dynamin-independent pathway that mediates uptake of GPI-anchored proteins, transmembrane proteins and bulk uptake of fluid-phase markers into GPI-anchored protein enriched early endosomal compartments (GEEC) derived from fusion of uncoated clathrin-independent carriers (CLICs) [45,46]. In CLIC/GEEC, cycling of Cdc42 activity is required for productive endocytosis. GTPases cycle between active GTP-bound and inactive guanosine diphosphate (GDP)-bound states that are regulated by GTPase activating proteins (GAPs), which stimulate GTP hydrolysis, and guanine nucleotide exchange factors (GEFs), which exchange GDP for GTP [47]. Both Cdc42 activation and inactivation are required for productive actin polymerization, and the small GTPase Arf1 recruits the Cdc42 GAP protein ARHGAP10/21 to mediate Cdc42 inactivation [42,43]. Activated Cdc42 also recruits the GAP GRAF1, which regulates Cdc42 and also has a BAR domain that is thought to contribute to membrane curvature and work together with forces generated by actin polymerization to drive membrane fission in a dynamin-independent manner [42,43,48].

### 2.3. Endocytic Trafficking to the Lysosome in Model Organisms

In yeast and mammals, endocytosed cargoes, whether they are derived from clathrin-mediated or clathrin-independent endocytosis, are trafficked sequentially through the Rab5 endosome, the Rab7 endosome and then the lysosome [42,49]. These trafficking events are coordinated by the Rab GTPases Rab5 and Rab7. Rab5 activity is maintained by its guanine exchange factor (GEF), RABGEF1/Rabex5, and recruits the class C core vacuole/endosome tethering factor (CORVET) complex, which drives homotypic fusion of Rab5 endosomes [49,50,51,52]. Then the Rab7 GEF, SAND1/Mon1-Ccz1 is recruited, leading to dissociation of RABGEF1/Rabex5 and Rab5 and recruitment of Rab7 [53,54]. Finally, CORVET (composed of Vps3, 8, 11, 16, 18, and 33) is converted to the homotypic fusion and vacuole protein sorting (HOPS) complex by switching out Vps3 and 8 for Vps39 and 41 [53,54,55].

Plants employ the same conserved machinery as yeast and mammalian cells for trafficking to the lysosome but have several distinguishing features. First, plants deliver endocytosed cargoes initially to the trans Golgi network (TGN), and emerging evidence suggests that cargoes may be delivered to the plant vacuole by at least three different pathways [56,57]. The SAND1/Mon1-Ccz1 complex acts as a Rab7 GEF and mediates conversion from Rab5 to Rab7 on endosomes in a Rab5 and Rab7-dependent pathway similar to yeast and mammalian cells. However, plants also have pathways that are Rab5- and Rab7-independent and Rab5-dependent but Rab7-independent [57,58]. At least some Rab5 endosomes are derived by maturation of the TGN [59]. Homologs of CORVET and HOPS are also present in plants and promote trafficking to the plant vacuole [60]. Studies investigating shared core CORVET/HOPS subunits, support a role in trafficking to the plant vacuole. Vps16 interacts with Vps11 and Vps33 at late endosomal and lysosomal membranes and regulates biogenesis and fusion of the plant vacuole [61,62,63]. However, a recent study suggests that CORVET and HOPS do not act sequentially in plants but regulate distinct trafficking pathways to the plant vacuole. Plant CORVET subunit Vps3 interacts with Rab5 and is thought to mediate fusion of endosomes to the plant vacuole in the Rab5-dependent, Rab7-independent pathway, while HOPS subunits Vps39 and Vps41 interact with Rab7 and regulate homotypic fusion of vacuoles for the Rab5- and Rab7-dependent pathway [64,65].

## 3. Mechanisms of Endocytosis in *Plasmodium* spp.

Similar to *Toxoplasma gondii*, *Plasmodium* spp. parasites also replicate in a PV during acute infection, but they do so by infecting red blood cells (RBCs) and consuming RBC cytosol from within this niche. Electron microscopy studies show that RBC cytosol is taken up across the PVM and parasite plasma membrane simultaneously [66,67]. The PVM extends in toward a mouth-like structure at the parasite plasma membrane called the cytostome. Subsequent pinching off of both membranes results in a bite of the RBC cytosol encased in two layers of membrane derived from parasite plasma membrane and PVM. 

Most attempts to define the molecular mechanisms involved in host cytosol uptake and delivery to the digestive vacuole have included small-molecule inhibitors. A role for dynamin was proposed after inhibition of *Plasmodium falciparum* dynamin-like protein 1 (PfDYN1) with the small-molecule dynamin inhibitor dynasore led to a reduction in the amount of hemoglobin internalized by the parasite [68]. Efforts to deplete the parasites of PfDYN1 to further elucidate its role in the uptake of host cytosol have not been fruitful, and analysis of its localization did not identify PfDYN1 at the cystosomal collar where actin was observed in some cases [69,70].

A study using actin inhibitors with different mechanisms of action, cytochalasin D (CytoD) and jasplakinolide (JAS), reported a decrease in the number of cystostomes following a short treatment [69]. Treatment also led to an increase in double membrane vesicles that were designated as part of a convoluted cystostome and further identified as cytostomal sections. This is comparable with the higher number of hemoglobin containing vesicles observed by other groups following actin inhibition [70,71]. Transport of hemoglobin to the food vacuole (FV) is negatively affected by actin polymerization, as observed by the inhibition of actin depolymerization using JAS and was further shown to be dependent on the actin-myosin motor [69,70]. Although these studies suggest a role for actin dynamics in endocytosis by malaria parasites in a manner akin to yeast [29,72], the precise role of actin in this process remain to be determined.

Rab5a was among the first proteins to be implicated in hemoglobin uptake by *Plasmodium falciparum* through genetic modification, in this case via expression of a constitutively active mutant [73]. However, more recent work using conditional mislocalization or conditional knockout of Rab5a resulted in growth arrest in the schizont stage [74]. The same study showed that conditional mislocalization of a Mon1 protein (a putative Rab7 GEF) was also growth arrested at the schizont stage. These findings are inconsistent with PfRab5a or PfMon1 playing a role in hemoglobin uptake, which principally occurs in the trophozoite stage. The first definitive genetic evidence of a protein’s role in hemoglobin uptake was reported for the vacuolar sorting protein 45, PfVps45. PfVps45 is a member of the Sec1/Munc-18 family, which in model systems is critical for activating SNAP receptor (SNARE) complexes for vesicular fusion. Conditional inactivation of PfVps45 led to an increase in vesicles containing host cytosolic material, thus impairing delivery of endocytosed material to the food vacuole [75]. Recent studies have provided additional insight into the molecular mechanisms involved in this process. The localization of the Kelch domain-containing protein (K13) at the cell periphery, forming a ring-like structure [76] and interacting with EPS15 have led to the identification of an endocytic compartment complex required for the uptake of hemoglobin [77]. Interestingly, the AP-2μ subunit, but not clathrin, localized with the K13-EPS15 complex, suggesting a potential role for clathrin-independent endocytosis that still requires the AP-2 complex. Several additional proteins were identified as interactors of K13 based on proximity biotinylation. Disruption of these interactors also blocked hemoglobin uptake, suggesting that the K13-EPS15 complex plays a crucial role in the early stages of uptake, possible during the formation of endocytic vesicles. Further in-depth characterization of the K13-EPS15 complex components should provide important new insight into endocytosis at the cytostome.

## 4. Endocytosis and Endocytic Trafficking in *Toxoplasma gondii*

### 4.1. Endocytic Trafficking across the PVM and Parasite Plasma Membrane

For the *Toxoplasma gondii* ingestion pathway, host cytosol must be taken up across the PVM and parasite plasma membrane before trafficking to the VAC, where it is degraded by cathepsin protease L (CPL) and other lysosomal hydrolases [11] (Figure 2). To track this process, fluorescent proteins such as GFP or mCherry are expressed in the host cytosol as non-specific tracers that can be visualized by microscopy. How these trafficking events are accomplished is not well established, but life inside a PV adds an extra layer of complexity. Possible mechanisms for endocytosis of host cytosolic fluorescent proteins into *Toxoplasma gondii* will involve trafficking across both the PVM and parasite plasma membrane to enter the parasite rather than just the plasma membrane in model organisms. 

Similar to *Plasmodium* spp., *Toxoplasma gondii* has a mouth-like structure at the plasma membrane called the micropore, which was proposed to be a site of endocytosis in *Toxoplasma gondii* [9]. Electron microscopy studies show vesicles occupying the micropore that were proposed to be derived from the PVM and thus would contain material derived from the host cytosol [9]. However, it is impossible to say whether the vesicle-like structures in the *Toxoplasma gondii* micropore are free vesicles or continuous with the PVM, or whether or being trafficked in or out of the parasite. Further, although micropores are present in acute and chronic stages of infection, micropores occupied by these vesicle-like structures have only been reported in the chronic cyst stage [9]. *Toxoplasma gondii* and *Plasmodium* spp. might employ similar mechanisms for endocytic trafficking across the PVM and parasite plasma membrane, but currently there is limited evidence to support a common mechanism. Further, putative endocytic vesicles have been observed in *Toxoplasma gondii* at sites that are distinct from the micropore upon treatment of infected cells with excess lipids [78] or tracking of endocytosis in extracellular parasites [79]. Such sites lack the hallmark collar of the micropore and are often seen in a more posterior site than the micropore. These observations raise the possibility of multiple and mechanistically distinctive sites of endocytosis in *Toxoplasma gondii*.

### 4.2. Components of Clathrin-Mediated Endocytosis in Toxoplasma gondii

Clathrin-mediated endocytosis is a major route of endocytic uptake that is thought to be used by all eukaryotic organisms [1]. Although it has not been directly tested, there is some evidence to support that *Toxoplasma gondii* undergoes clathrin-mediated endocytosis. The micropore sometimes appears to have a proteinaceous coat detectable by electron microscopy [9]. There are two forms of endocytosis with characteristic coats visible by electron microscopy, clathrin-mediated endocytosis and caveolar endocytosis. *Toxoplasma gondii* does not express caveolin/cavin proteins required for caveolar endocytosis, and thus it is reasonable to consider clathrin as a prime candidate for a coat protein [80]. Consistent with a potential role for clathrin, many classical endocytic players are conserved in *Toxoplasma gondii* (e.g., dynamin-related proteins, epsin-like proteins, and actin) [8,81,82,83,84,85].

However, neither clathrin nor actin, dynamin-related proteins, or epsin-like proteins have been localized to sites of endocytosis at the parasite plasma membrane [8,81,82,83,84,85]. These proteins have been implicated in other roles including parasite motility and organelle dynamics (actin), maintenance of the chloroplast-like organelle called the apicoplast (the dynamin-related protein, DrpA), exocytic trafficking to the parasite secretory organelles called micronemes and rhoptries (clathrin, DrpB and potentially the epsin-like protein TgEpsL) [8,81,82,83,84,85] or daughter cell formation and organelle maintenance including fission of the parasite’s single mitochondrion (DrpC) [81,86]. Further, a bioinformatic search for homologs of genes in the yeast clathrin interactome revealed a paucity of effectors required for clathrin-mediated endocytosis [81]. While Pieperhoff et al. [81] found homologs for the effectors mentioned above, the scaffolding protein EPS15 and the clathrin adaptor complex AP-2, they did not find homologs for other crucial players in yeast and mammalian cells such as the nucleator intersectin, adaptors AP180 or epsin, actin polymerizing and branching proteins N-WASP and Arp2/3, Hip1R, which links actin to the clathrin coat, or uncoating proteins related to auxillin [1,27,81]. 

Regardless, missing key effectors may not be problematic. It is possible that *Toxoplasma gondii* does not require components of the canonical clathrin-mediated endocytosis machinery or may have evolved novel effector proteins. Protein BLAST (basic local alignment search tool) searches did not reveal obvious homologs of any TPLATE complex members in the *Toxoplasma gondii* genome, but other yet to be discovered parasite-specific effectors could exist and coordinate clathrin-mediated endocytosis. For example, *Toxoplasma gondii* does encode two hypothetical BAR domain proteins (TGME49_259720 and TGME49_320760) and a hypothetical ENTH-domain containing protein (TGME49_216030) that could participate in membrane bending or nucleation or endocytosis like FCHO1/2 (Table 1).

It is important to note that while many of the canonical clathrin-mediated endocytosis effectors have established roles in exocytic trafficking in *Toxoplasma gondii*, only DrpB has been functionally tested in the context of host protein endocytosis. DrpB is not required for *Toxoplasma gondii* endocytosis, suggesting that it is dedicated to exocytic trafficking [23]. However, this should not be interpreted to mean that *Toxoplasma gondii* endocytosis is independent of dynamin. As noted above, *Toxoplasma gondii* expresses two other dynamin-related proteins, DrpA and DrpC. DrpA is responsible for the maintenance of the apicoplast [84]. DrpC localizes to cytoplasmic puncta and is associated with formation of daughter parasites and mitochondria constriction; however, it physically interacts with components of the AP-2 complex, a homolog of intersectin-1, and a homolog of the malaria K13 protein [86]. How its subcellular localization is tied to its physical interaction with components of endocytosis and vesicular trafficking remains unknown. Since the roles of DrpA and DrpC have not been tested in the context of host protein endocytosis, it is still possible that a dynamin-related protein could drive endocytic vesicle fission. Regardless, testing the role of clathrin itself in host protein endocytosis will ultimately determine whether clathrin-mediated endocytosis exists in *Toxoplasma gondii*. This should be a priority in the field. 

### 4.3. Components of Clathrin-Independent Endocytosis into Toxoplasma gondii

It is also possible that ingestion of host proteins by *Toxoplasma gondii* could resemble clathrin-independent mechanisms and be coordinated by actin, regulatory Rho family GTPases, and/or BAR-domain containing proteins [42,43,91]. In the absence of obvious N-WASP and Arp2/3 homologs, actin could be nucleated by formins similar to the Rho1-dependent pathway in yeast. This pathway is dependent on Rho1 and formin, but independent of clathrin and Arp2/3 [92]. *Toxoplasma gondii* expresses three formin proteins capable of polymerizing actin (Frm1, Frm2 and Frm3). Whereas *Toxoplasma gondii* Frm1 and Frm2 promote parasite motility during invasion, Frm3 is not required for invasion and localizes to a discrete, unidentified structure at the apical end of the parasite [89,93]. *Toxoplasma gondii* also has two hypothetical Rho-like GTPases and a hypothetical RhoGAP-like protein, TGME49_267045, TGME49_249170 and TGME49_233300, respectively (Table 1). Alternatively, proteins with N-BAR or ENTH domains that have amphipathic helices may act independently in membrane scission. The N-BAR domains of endophilin and amphiphysin and the ENTH domain of epsin have amphipathic helices and are capable of vesiculating liposomes in vitro [94,95]. Further, overexpression of epsin in mammalian cells can rescue scission of clathrin-coated vesicles when dynamin is knocked down [95].

### 4.4. Potential Contribution of Parasite Proteins Associated with the PVM and PV Lumen

While parasite cytosolic proteins discussed above putatively coordinate invagination and scission of the parasite plasma membrane, there is no evidence that such proteins are secreted. Thus, other factors must be involved in creating vesicles at the PVM. Parasite proteins secreted into the PV lumen from secretory organelles called the dense granules modify the PV and are thought to facilitate acquisition of nutrients from the host cell. GRA2, 4 and 6 promote formation of a system of membrane tubules invaginated from the PVM called the intravacuolar network (IVN), which is thought to act like villi, increasing surface area of the PVM for maximum nutrient exchange [96,97,98]. GRA7 has been implicated in scavenging cholesterol from the host cell via structures called Host Organelle Sequestering Tubulo-structures (H.O.S.T.s) [20]. In H.O.S.T.s, host lysosomes are recruited to the PV on microtubules, engulfed in PV membrane invaginations and pinched off, presumably by GRA7, to form lysosome-containing vesicles in the PV lumen. The ability of *Toxoplasma gondii* to ingest host proteins is reduced by 40% in parasites lacking GRA2 and unaffected in the absence of GRA7 [11]. This precludes a H.O.S.T.-like mechanism and suggests that the IVN contributes to host protein ingestion. In addition to its suggested role in nutrient exchange, the IVN also acts as a structural component to organize replicated parasites into a characteristic rosette pattern within the PV [99]. It remains to be determined whether the IVN plays a direct role in the ingestion pathway by acting as vesicular conduit or an indirect role as a structural organizer of parasites in the PV.

### 4.5. Potential Contribution of Host Proteins for Trafficking across the PVM

Host proteins also have the potential to contribute to *Toxoplasma gondii* endocytosis across the PVM. The host Endosomal Sorting Complex Required for Transport (ESCRT) is a rational candidate for several reasons. First, ESCRT protein complexes (ESCRT-0, -I, -II, -III and the ATPase Vps4) are expressed in the host cell cytosol and are sequentially recruited to endosomal compartments to invaginate (ESCRT 0–II) and pinch off endosomal membranes (Vps4) to release vesicles into the lumen of the endosomal compartment [100]. The topology of this reaction matches the requirement for *Toxoplasma gondii* endocytosis, membrane bending and scission away from the cytosol to create vesicles in the PV lumen. 

Second, a recent study found that host ESCRT machinery is hijacked by *Toxoplasma gondii* for invasion. PT/SAP, YPXL, and PPXY amino acid motifs were recently found in the *Toxoplasma gondii* rhoptry neck proteins RON2, 4 and 5 [101]. These same motifs are found in the Gag proteins of retroviruses such as Human Immunodeficiency Virus (HIV) and Rous Sarcoma Virus and are well-documented motifs for recruitment of the ESCRT complex for budding from the plasma membrane [102,103]. PT/SAP and YPXL motifs in the cytosolic portion of HIV Gag directly recruit Tsg101 of ESCRT-I and the ESCRT-associated protein ALIX [103]. ALIX binds to HIV Gag, Tsg101 and CHMP4 of ESCRT-III, and facilitates recruitment of Vps4 to mediate scission of HIV particles out of the cell [103]. PPXY motifs in Rous Sarcoma Virus Gag recruit Nedd4 family ubiquitin ligases, which may ubiquitinate Gag or recruit arrestin-related trafficking proteins (ARTs) to provide a scaffold for ESCRT assembly at the viral budding site [100,103,104,105,106,107]. Similarly, these motifs in RON4 and 5 facilitated recruitment of ALIX and Tsg101 to the moving junction for host cell invasion [101]. PX(P/A)XPR motifs were also identified in RON2, 4 and 5 and contributed to recruitment of the ALIX binding partner CIN85 and its homolog CD2AP to the moving junction [101]. 

Interestingly, host protein endocytosis is initiated during or immediately following invasion and is active through at least the first 6 h post-invasion [23]. RON2, 4 and 5 are in the moving junction complex during host cell invasion, and at least part of this complex remains on the PVM for several hours following invasion, as evidenced by staining of RON4 and RON5 [108,109]. It will be interesting to know if RON4 and 5 remain competent to recruit ESCRT to the PVM following invasion and if ESCRT or other host proteins contribute to *Toxoplasma gondii* endocytosis.

### 4.6. Does Endocytic Trafficking in Toxoplasma gondii Resemble Yeast/Mammals or Plants?

*Toxoplasma gondii* belongs to the phylum Apicomplexa, which has evolutionary origins that are thought to predate the split between plants and animals and share features of both [110]. In relation to plants in particular, *Toxoplasma gondii* possesses a vacuolar pyrophosphatase proton pump (VP1) and a Rab5 isoform (Rab5B) specific to plants that are associated with its endolysosomal system [2,5]. This suggests endocytic trafficking to the VAC could resemble either yeast/mammals or plants. A recent study demonstrated that endocytic trafficking proceeds through the ELCs, but whether endocytic trafficking also proceeds through the TGN as in plants remains unclear, since current markers used to identify the TGN also label the ELCs to some extent [23]. That fluorescent phospholipids endocytosed by extracellular parasites colocalized with Vps53, a putative TGN marker, provides further support for the potential involvement of the parasite TGN as a staging site for endocytic material [79]. Plant-like features of *Toxoplasma gondii* endocytosis should be of particular interest for future work given their therapeutic potential and identifying specific and validated markers of the TGN will be a crucial first step.

### 4.7. Endocytic Trafficking to the VAC Intersects with Exocytic Trafficking to the Micronemes

*Toxoplasma gondii* also expresses key players for trafficking of endocytosed material to the lysosome including three Rab5 proteins, the Rab5 GEF Vps9, Rab7, the Rab7 GEFs Mon1 and Ccz1, the core subunits shared by the CORVET and HOPS complexes (Vps11, Vps16, Vps18, Vps33) and the HOPS-specific subunits Vps39 and Vps41 [5,6,111]. Based on model systems, trafficking to the VAC would be predicted to involve a Rab5 to Rab7 switch mediated by recruitment of the Rab7 GEF SAND1/Mon1-Ccz1 and conversion of the CORVET complex to the HOPS complex [26,53,54]. However, exocytic trafficking to the parasite’s unique secretory organelles called the micronemes and rhoptries proceeds through the TGN and ELCs and requires Rab5A, Rab5C, Vps9, and the HOPS complex [6]. This could indicate that these proteins, which are typically involved in endocytic trafficking, have been uniquely adapted for exocytic trafficking and secretion. Alternatively, endocytic and exocytic trafficking could share at least some common trafficking mechanisms. Recent evidence suggests that endocytic trafficking of ingested host proteins intersects with exocytic trafficking of microneme proteins but not rhoptry proteins [23]. Similar findings of an intersection of the endocytic and exocytic systems were reported from tracing the uptake and trafficking of fluorescent phospholipids and nano-gold particles in extracellular parasites [79]. The merging of endocytic and exocytic trafficking is intriguing given that ingested host proteins are sent to the VAC for degradation whereas microneme proteins must arrive intact at the microneme organelles to mediate their essential roles in parasite invasion and egress [3,5,6,8,81,111,112]. Therefore, in addition to possibly sharing trafficking mechanisms, especially where trafficking is merged, there will need to be distinct sorting mechanisms to ensure that endocytic and exocytic cargoes are sorted to their appropriate destinations.

Sorting of microneme and rhoptry proteins is partially understood. The sorting receptor sortillin (TgSORTLR) and transmembrane microneme and rhoptry proteins bind to soluble microneme and rhoptry proteins in the TGN [113,114,115]. The clathrin adaptor protein complex AP-1 is thought to recognize tyrosine-based and acidic sorting signals in the cytoplasmic portion of TgSORTLR or transmembrane microneme and rhoptry proteins [4,8]. AP-1 also interacts with the epsin-like protein TgEpsL, and together they are thought to recruit clathrin for sorting from the TGN to the ELCs via clathrin-coated vesicles that are pinched off by DrpB [8,86]. How microneme and rhoptry proteins are sorted from ELCs is not known. 

Whether ingested host proteins utilize sorting mechanisms that overlap with those of microneme and rhoptry is also unknown and should be a focus of future studies. Further, it is important to remember that if the *Plasmodium* model of endocytosis is correct, then what is actually being trafficked to the VAC are host protein-containing vesicles. These endocytosed vesicles could be delivered to the VAC for degradation via a bulk flow pathway as demonstrated for endocytosed red fluorescent protein, RFP, delivered to the TGN in plants without the help of any sorting signals or sorting receptors [49,116,117]. On the other hand, sorting receptors could recognize parasite proteins embedded in the PVM-derived vesicle membrane. Recognition by such receptors could begin as early as the plasma membrane and could guide trafficking anywhere along the pathway to the VAC.

It will also be interesting to understand other mechanisms that distinguish endocytic and exocytic trafficking in *Toxoplasma gondii*. That disruption of TgVps45 affects the trafficking of ingested host proteins to the VAC, but not the biogenesis or discharge of microneme or rhoptry proteins [90], suggests distinct mechanisms. Further, DrpB is required for trafficking of microneme and rhoptry proteins from the TGN to their respective organelles, but DrpB is not required for trafficking of ingested host proteins to the VAC [23]. The roles of Rab5 proteins, Rab7, their associated GEFs, CORVET and HOPS in *Toxoplasma gondii* endocytosis have not been tested. Like plants and *Plasmodium* spp., *Toxoplasma gondii* expresses two C-terminally geranylated, conventional Rab5 isoforms, Rab5A and Rab5C, and the aforementioned plant-specific isoform that is N-terminally myristoylated, Rab5B [5,118,119]. The conventional Rab5A and Rab5C show nearly perfect colocalization on endosomal membranes, and expression of dominant negative mutants of Rab5A, Rab5C and the Rab5A GEF Vps9 disrupts trafficking of a subset of microneme and rhoptry proteins to their respective organelles [5,111]. The plant-like Rab5B colocalizes partially with RabA and Rab5C and also to the parasite plasma membrane [5]. Rab5B has no defined function since expression of dominant negative Rab5B had no effect on exocytic trafficking to the microneme and rhoptry organelles [5]. Similarly, expression of dominant negative Rab7 also did not interrupt microneme and rhoptry biogenesis, suggesting that Rab7 is not required for exocytic trafficking in *Toxoplasma gondii*. Yet, the putative Rab7 GEF Mon1 is required for the biogenesis of micronemes and rhoptries [5,6], thus clouding the role of Rab7 in exocytic trafficking. The possibility that Rab5A and Rab5C are dedicated to exocytic trafficking and Rab5B, Rab7, and a second putative Rab7 GEF, Ccz1, are devoted to endocytic trafficking is worthy of additional investigation.

### 4.8. The Function and Importance of Toxoplasma gondii Endocytosis

Although the exact function and importance of *Toxoplasma gondii* ingestion are not known, there is reason to believe that endocytosis could be central to parasite survival. Endocytosis is essential for nutrient acquisition, intracellular signaling, plasma membrane homeostasis, and receptor turnover in eukaryotes [1]. Further, host cytosol ingestion in *Plasmodium* spp. parasites is essential for acquisition of amino acids [120]. Targeting this pathway has been a successful therapeutic strategy [121,122]. 

Beyond nutrient acquisition from the host cell, endocytosis could be important for cell signaling, membrane turnover, or specifically in the case of *Toxoplasma gondii* evading the immune response or sensing the host cell environment [1]. Accordingly, the virulence defect of CPL knockout parasites is completely rescued in the absence of interferon gamma receptor, suggesting that protein degradation in the VAC is somehow involved in immune evasion [11]. CPL was also recently identified in a genetic screen as conferring fitness in interferon gamma activated macrophages [123]. As for sensing, although ingestion of host cytosolic mCherry across the PVM is expected to occur by a bulk flow-like mechanism, the parasite could have receptors on the PVM that bind to and internalize components of cell intrinsic defense systems as an immune evasion strategy or it could use PVM receptors as a sensory system to monitor the health of the cells it infects. 

Finally, although host cytosol ingestion has been recently reported to also occur in chronic stage *Toxoplasma gondii* bradyzoites [124], the precise role of this pathway in bradyzoite biology remains to be determined. Parasites lacking CPL can differentiate into cysts normally, but eventually accumulate undigested autophagosomes and die [125]. Whether autophagosomes accumulate solely because they are not being turned over or are induced in response to starvation for host-derived nutrients such as amino acids, including those derived from the degradation of ingested host cytosolic proteins, is unclear. Future dissection of these possibilities will help clarify how the parasite uses its endolysosomal system to persist indefinitely in its hosts.

## 5. Conclusions

How, when, and why *Toxoplasma gondii* ingests proteins from the host cell cytosol largely remain a mystery. Endocytosis in model systems was heavily relied upon for envisioning how endocytosis works in *Toxoplasma gondii* due to the infancy of the field, but what is most exciting are the peculiarities of *Toxoplasma gondii* and potential parasite-specific aspects that cannot be predicted. The paucity of conserved endocytic players and the fact that *Toxoplasma gondii* performs endocytosis across two membranes presents an exciting opportunity to discover potentially novel aspects of endocytosis for eukaryotes. *Toxoplasma gondii* may also serve as a useful model of endocytosis in other parasitic pathogens. The analogous endocytic pathway for ingestion of red blood cell cytosol in *Plasmodium* spp. has been a successful therapeutic strategy, but drug resistance to nearly every antimalarial treatment is emerging and yet there remains very little known about how this pathway works [120,121,122,126]. It is also not known whether other apicomplexan parasites have analogous ingestion pathways, including *Cryptosporidium* which causes serious diarrheal illness that is often not curable in immunocompromised individuals [127]. Although nothing is known about ingestion by *Toxoplasma gondii* sexual stages, *Eimeria* macrogametes were reported to internalize nanoparticles [128], and thus further interrogation of apicomplexan sexual stages is warranted. Future studies of *Toxoplasma gondii* endocytosis have the potential to deliver exciting and impactful results that could help explain the remarkable success and promiscuity of this intracellular pathogen.

## Figures and Tables

**Figure 1 life-11-00084-f001:**
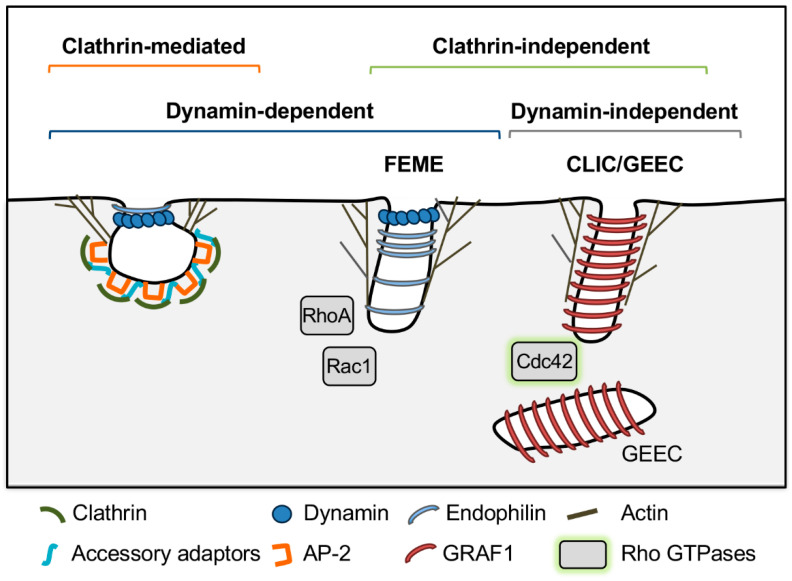
Mechanisms of endocytosis in model systems. Note that these models are simplified from the text to depict key differentiating features of each pathway. See text for full description of endocytic mechanisms. Fast endophilin-mediated endocytosis (FEME), clathrin-independent carriers/glycosylphosphatidylinositol (GPI)-anchored protein enriched early endosomal compartments (CLIC/GEEC). Caveolin-mediated endocytosis was omitted because it is not relevant to *Toxoplasma gondii* and other apicomplexan parasites, which lack caveolin.

**Figure 2 life-11-00084-f002:**
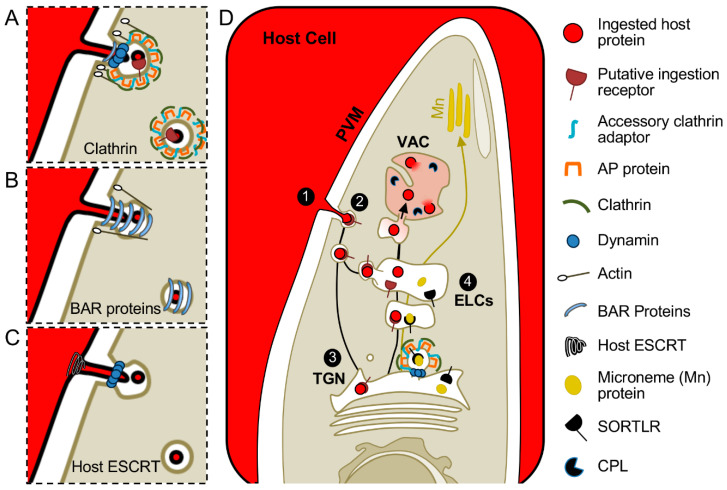
Hypothetical models for *Toxoplasma gondii* endocytosis and trafficking of ingested protein. Ingested proteins are proposed to traverse the PVM in internally budding vesicles and enter the parasite at a discrete site. Mechanisms could resemble model systems and depend on coat proteins, dynamin, and actin. Examples of (**A**) clathrin-mediated or (**B**) clathrin-independent processes, represented by CLIC-GEEC, are depicted. (**C**) Scission of endocytic vesicles may also require host proteins for pinching of the PVM as depicted by possible recruitment of host Endosomal Sorting Complex Required for Transport (ESCRT) machinery and garroting at the parasite plasma membrane by a dynamin-related protein. (**D**) After scission from the PVM (step 1) and entering the parasite (step 2), guided by a putative ingestion receptor, the ingested material is proposed to traverse the TGN (step 3) before trafficking through the ELCs (step 4) en route to the VAC for degradation by proteases including CPL. Mn, microneme; Rop, rhoptry.

**Table 1 life-11-00084-t001:** *Toxoplasma gondii* endocytic proteins.

GeneID	Protein Name	Domain/Complex	Known or Prospective Function	Phenotype Score [87]
TGME49_290950	Clathrin	Heavy chain	Interacts with AP-1 mu1 at TGN [8], promotes microneme and rhoptry trafficking at the TGN [81]	−4.77
TGME49_221940	AP2	Alpha subunit	Clathrin adaptor for endocytosis? [88]	−4.32
TGME49_240870		Beta2 subunit		−2.83
TGME49_230920		Mu2 subunit		−1.60
TGME49_313450		sigma-1 subunit		−2.88
TGME49_320760	Hypothetical	BAR/IRSp53 and MIM homology domain (IMD)-like domain	Promote membrane curvature?	0.56
TGME49_259720	Hypothetical	BAR/IMD-like domain	Promote membrane curvature?	−4.55
TGME49_216030	Hypothetical	ENTH/ Vps-27, Hrs and STAM (VHS) domain	Promote membrane curvature, clathrin adaptor protein?	−0.67
TGME49_227800	TgEPS15	Eps15 homology domain	Clathrin-mediated endocytosis nucleation/scaffold protein?	−3.19
TGME49_214180	TgEpsL	ENTH domain	Interacts with TgAP-1 at TGN but likely not AP-2, formation of clathrin coated vesicles in secretory trafficking [8]	−0.52
TGME49_213370	Frm3	Actin binding	Actin nucleation during endocytosis? [89]	−2.79
TGME49_267045	Hypothetical	Ras superfamily/Rho GTPase	Regulation of actin during endocytosis?	0.81
TGME49_249170	Hypothetical	Ras superfamily/Rho GTPase	Regulation of actin during endocytosis?	−0.79
TGME49_233300	Hypothetical	Rho GTPase activating protein (RhoGAP)	Regulation of actin during endocytosis?	1.52
TGME49_321620	DrpB	GTPase	Microneme and rhoptry trafficking at the TGN	−4.91
TGME49_270690	DrpC	GTPase	Mitochondrial maintenance? [81]	−4.54
TGME49_267810	Rab5A	GTPase	Microneme and rhoptry trafficking at the TGN/ELCs [5]	−4.48
TGME49_207460	Rab5B	GTPase	Undefined [5], recycling to plasma membrane?	0.55/−1.35
TGME49_219720	Rab5C	GTPase	Microneme and rhoptry trafficking at the TGN/ELCs [5]	−4.24
TGME49_248880	Rab7	GTPase	Undefined [5], fusion with the lysosome?	−2.67
TGME49_230220	Vps11	CORVET/HOPS	Exocytic trafficking to the micronemes and rhoptries [6], biogenesis of the VAC [6], endocytic trafficking through Rab5 and Rab7 compartments to the VAC?	−4.09
TGME49_320670	Vps16	CORVET/HOPS	Exocytic trafficking to the micronemes and rhoptries? Endocytic trafficking through Rab5 and Rab7 compartments to the VAC?	−4.82
TGME49_289730	Vps18	CORVET/HOPS	Exocytic trafficking to the micronemes and rhoptries [6], endocytic trafficking through Rab5 and Rab7 compartments to the VAC?	−3.24
TGME49_295000	Vps33	CORVET/HOPS	Exocytic trafficking to the micronemes and rhoptries? Endocytic trafficking through Rab5 and Rab7 compartments to the VAC?	−4.78
TGME49_224270	Vps41	HOPS	Exocytic trafficking to the micronemes and rhoptries? Endocytic trafficking through Rab5 and Rab7 compartments to the VAC?	−3.75
TGME49_315530	Vps39	HOPS	Exocytic trafficking to the micronemes and rhoptries [6], endocytic trafficking through Rab5 and Rab7 compartments to the VAC?	−4.18
TGME49_291120	Mon1	SAND1/Mon1-Ccz1	Rab7 GEF? Exocytic trafficking to the micronemes and rhoptries [6], endocytic trafficking through Rab5 and Rab7 compartments to the VAC?	−4.42
TGME49_207960	Ccz1	SAND1/Mon1-Ccz1	Rab7Gef? Exocytic trafficking to the micronemes and rhoptries? Endocytic trafficking through Rab5 and Rab7 compartments to the VAC?	−3.96
TGME49_271060	Vps45	Sec1 family protein	Vesicular trafficking of endocytosed material to the VAC [90]	−5.04
TGME49_224710	VSR1	PA domain	Undefined, cargo recognition for sorting to the VAC?	−0.90
TGME49_312860	VSR2	PA domain	Undefined cargo recognition for sorting to the VAC?	−0.09

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
