# Peer review of "Emerging Mechanisms of Endocytosis in Toxoplasma gondii"

_life, 2021, doi:10.3390/life11020084_

Round 1
Reviewer 1 Report
McGovern et al. review clathrin-dependent and -independent mechanisms briefly in model organisms, then discuss what is known in Plasmodium and then describe current knowledge in Toxoplasma gondii. This review is a good mix of reviewing studies, hypothesis and potential future questions that could be addressed. Particularly useful is figure 1 as well as the table that lists endocytic proteins.
Major issues:
- Only recently. Spielmann et al. have published a review on endocytosis in T. gondii and Plasmodium which needs to be considered in this work.
- Paragraph line 190: Inhibitors of actin polymerization and depolymerization have similar effects on cytostome formation. The authors mention that the role of actin in endocytosis in not known. Can we speculate from the inhibitor studies that actin turnover is critical for endocytosis? Maybe the authors could briefly refer here to yeast (e.g. review: https://doi.org/10.1534/genetics.112.145540 and doi: 7554/eLife.52355)
- Figure 2D: Mn has to be described in the legend. Maybe draw a bit differently and either leave out the one rhoptry or label it too.
- L252: The sentence starting with “Its is tempting…” should be re-phrased so it becomes clearer what is meant (similar but not common mechanisms).
- L269: Please add something like “for a coat protein” after “primary candidate”.
- L331: Please correct first part of this sentence.
- L347: Please add “It” or something similar at the beginning of the sentence.
- This review is mainly based on tachyzoites, but chronic stages are mentioned. As the study of the cat stages is becoming a more emerging topic, maybe the authors could add a few sentences at least in the conclusion to this topic about this large knowledge gab. Frölich and Wallach (https://doi.org/10.1038/srep29030) have done experiments using macrogametocytes from Eimeria maxima. This paper could be cited too, especially as they have done experiments using cytochalasin D and observed inhibited uptake of nanoparticles.
Minor issues:
- Line 30: Please replace „wasn’t“ with “was not”
- Line 104: Please omit “a” in “a one study”
- L158: Please expand TGN for the first time (maybe also check all the abbreviations?).
- L187: Please replace “haven’t” with “have not”. Maybe delete “and” and start new sentence.
- L189: Please replace “was” with “has”.
- L199: Can we really talk about the mechanism of actin? Should it not be rather the role of actin of the mechanism of endocytosis in which actin plays a role?
- L223: I would suggest re-phrasing the first part of the sentence “In T. gondii ingestion,…
- L234: Please add “the” between “traverse” and “PVM”.
- L240: Please delete one “the” at the end of the line.
- L245: Keep to past tense.
- L266: You mention two forms of endocytosis with coats. In which organisms? In most?
- L453: The sentence “Rab5B has no defined function, but expression of dominant negative Rab5B had no effect on exocytic trafficking to the microneme and rhoptry organelles [5].” is not clear to me. Maybe replace “but” with “and additionally”?
Reviewer 2 Report
In this manuscript, the authors review what is known with regards to endocytosis in the apicomplexan parasite Toxoplasma gondii. They start with a brief introduction as to why the presence of endocytosis was only recently uncovered in T. gondii. The following section summarizes endocytosis and endolysosomal trafficking in model organisms and highlight differences between yeast/mammalian cells and plants. The authors then discuss what is known with regards to endocytosis in the malaria parasite Plasmodium, a related apicomplexan, after which they move on to T. gondii.
I enjoyed reading the review. It is nicely written. The Carruthers’ lab has been critical in uncovering mechanisms of endocytosis in T. gondii so the manuscript is authoritative. The figures used are appropriate. A table containing T. gondii demonstrated and putative endocytic proteins is presented.
I only have minor comments:
-It would be good to separate the rows of Table 1 by lines as it can be hard to differentiate, especially for the Known or Prospective Function column.
-Line 461: Contrarily to what the authors claim, T. gondii possesses orthologues of both CORVET specific subunits (VPS3 and VPS8). See reference: Woo et al, eLife, 2015. “Chromerid genomes reveal the evolutionary path from photosynthetic algae to obligate intracellular parasites.”
Reviewer 3 Report
In this review McGovern et al summarise what is known about the mechanisms of endocytosis in Toxoplasma parasites. The review first describes endocytosis mechanisms known from model organisms, followed by a short summary of endocytosis in Plasmodium falciparum. The main part of the article then provides a detailed discussion of endocytosis in Toxoplasma gondii, including the evolutionary relationship of endocytic trafficking, connections and functions in respect to secretory trafficking and the possible relevance of endocytosis for parasite survival.
Endocytosis is an emerging field in Toxoplasma research and much about Apicomplexan endocytosis remains to be experimentally determined. The manuscript does justice to that fact by not only summarising what is experimentally known but also listing the proteins the parasite encodes that could be involved in endocytic processes (Table 1) and through a logical coverage of different options how host cell materials could be internalised into the parasite. This includes the more 'standard' clathrin dependent and independent endocytosis processes but also evidence and ideas for mechanisms that depend on the unique intracellular situation such as e.g. the possibility of vesicle-induction through host ESCORT components. The review also suggests targets for future research and together with the list of proteins that could be involved, is a good guide for the direction of future research.
Overall this is a very nice and interesting review from experts of this topic. I have only some minor comments, none critical, that might improve the manuscript.
- Line 104: One should be On; remove the 'a' after 'On the one hand'.
- CIM: I see the need to simplify the CIM mechanisms as they are often not well defined and less studied than CME. However, at least the caveolin pathway might be added. Although absent in apicomplexan parasites, it is mentioned later in the review and it would be helpful for the reader to know that it exists.
- Line 203: In this study Rab5a was analysed by both, conditional mislocaliation and diCre-based conditional KO (with the same result). It is buried in the supplementaries, but this study also conditionally inactivated PfSand1/Mon1. As this protein is discussed in this review more than once it might be useful to mention here that in P. falciparum its inactivation also showed a schizont phenotype, like Rab5a, indicating that it is not involved in endocytosis. In Table 1 there are two Sand1/Mon1 homologs listed for T. gondii. Could this be relevant in the context of secretory and endocytic functions in vesicle trafficking in T. gondii?
- Legend figure 2, insert 'the' before PVM and remove one of the 'the' before 'ingested material'
- Line 246 and model in Figure 2: Host cell cytosol filled vesicles occupying the micropore: I do not fully agree that this is akin to current models in P. falciparum, as in this organism the cytostome does not occupy vesicles: in malaria parasites the cytostome is bounded by both, the internalised plasma membrane and the PVM which appear as a double membrane. To my knowledge, smaller vesicles have never been described within the cytostome. What is the evidence in T. gondii that the micropore is a container of smaller host cell derived vesicles bounded by PVM (apart from the data in the cyst stage)? Couldn't the mechanism be akin to P. falciparum and either the entire structure pinch off or give rise to double membraned vesicles?
- Line 347: There is an 'It' missing after the full stop.
- Line 356: full stop missing
- Line 410: fix format of citation
- Line 442: for TgVPS45 references 4 and 8 are cited but they do not seem to analyse this protein. Was reference 88 meant, which does analyse VPS45? There is also an additional paper looking at VPS45 in T. gondii, although it did not analyse endocytosis (https://doi.org/10.1111/mmi.14411)
- Line 492-494: this is a complicated sentence. Maybe adding commas before 'including' and before 'is unclear' could help (or rephrasing).
- The entries in the properties column of the Table 1 are sometimes difficult to assign to the correct protein. Maybe the table can be reformatted to better separate the entries between the different rows. Please also fix the format of the citations in that column
